# Understanding Land in the Context of Large-Scale Land Acquisitions: A Brief History of Land in Economics

**Marcello De Maria** [1,2]

1  School of Agriculture, Policy and Development, University of Reading, Whiteknights Campus, Reading RG6 6AH, UK; m.demaria@pgr.reading.ac.uk
2  Data Analyst and Researcher-Land Portal Foundation, Postbus 716, 9700 AS Groningen, The Netherlands

**Abstract:** In economics, land has been traditionally assumed to be a fixed production factor, both in terms of quantity supplied and mobility, as opposed to capital and labor, which are usually considered to be mobile factors, at least to some extent. Yet, in the last decade, international investors have expressed an unexpected interest in farmland and in land-related investments, with the demand for land brusquely rising at an unprecedented pace. In spite of a fast-growing literature analyzing the variety of "*spaces*" affected by large-scale land acquisitions (LSLAs), the contemporary process of "*commodification*" of land embedded in this phenomenon has taken present day economists by surprise. This paper reviews the evolution over time of the concept of land in economics and it suggests how different aspects of this evolution are relevant to the understanding of contemporary LSLAs. Rather than presuming to analyze in a systematic and comprehensive manner the immense literature in land economics, this article investigates what makes land a peculiar and complex commodity. Indeed, different branches of economic thought, at different moments in time, pointed out that the location of land in space matters; that land is a living and fundamental component of the ecosystem; that it is a valuable economic asset, and yet, it is often hard to value it in pure monetary terms; eventually, that land is intrinsically connected to societies, cultural and spiritual identities, mores, and institutions. Through a brief history of the evolution of the concept of land in economics, this paper identifies four broad categories—namely, *space*, *economics*, *environment*, and *institutions*—that help understanding land as a peculiar good. These four elements characterize land as a commodity, as well as its peculiarities, and constitute the prerequisites of a conceptual framework for the analysis and the understanding of the forces at play in the contemporary wave of large-scale land acquisitions.

**Keywords:** large-scale land acquisitions; land grabbing; land investments; land market; land tenure; land economics; institutions; geography; environment

## 1. Introduction: Rethinking Land as a Commodity in the 21st Century

In a fast-changing, complex, and globalized world, assumptions, theories, and definitions that have been deemed to reflect well our world for long time, suddenly, need to be questioned, adapted, and updated. Indeed, in the last decade, the surge of large-scale land acquisitions (LSLAs)—also commonly referred as "*land grabbing*" by those who prefer to focus more on its controversial aspects—suggests that it is time for rethinking the role of land in economics, as well as in other sciences.

Researchers from different fields immediately recognized the strategic importance of LSLAs [1–6]. A recent but rapidly growing literature addressed a wide range of issues both from qualitative and quantitative perspectives, highlighting the variety of "*spaces*" affected by transnational land

deals, as well as the multidimensional and intertwined nature of the phenomenon. Climate change, food security, food sovereignty, environmental sustainability, land tenure security, land–energy nexus, and development issues, are some of the many aspects which this literature connected to LSLAs.

However, it seems that to some extent economists had been taken by surprise and a crucial part of the story is still, surprisingly, missing. The LSLAs phenomenon shows the existence of an international market for land, which implies that the process of "*commodification*" has started on a global and transnational scale for this resource. Nevertheless, the economic science seems to be not equipped yet with a conceptual framework allowing for the full understanding of the mechanisms and the forces at play in such a newly born international market for land.

For this reason, the different sections of this article aim—each one with a different specific focus—at filling this gap. While arguing that a new holistic conceptual framework for land is required in order to reflect the multifaceted features of the current LSLAs, I also endorse in this paper the idea that looking at the history of different branches and moments of the economic thought and other parallel disciplines, can offer invaluable insights in deciphering the present—and the future—of the most recent rush for land in human history.

Yet, before starting with a brief—and necessarily incomplete—history of land economics, it is important to discuss the main features characterizing the LSLAs phenomenon itself, and how they concur in justifying the need for a reconceptualization of land. Hence, in the next section I revise the many faces of LSLAs, highlighting through the relevant literature the main figures and the leading—and sometimes conflicting—narratives around its nature and consequences. I then describe the methodological approach, while the rest of this paper analyses the four crucial aspects that need to be considered in land-related studies and contribute to defining a comprehensive conceptualization of land at present day—namely *economics*, *space*, *environment*, and *institutions*. The last section concludes offering final remarks.

## 2. Large-Scale Land Acquisitions

In the last decade, international investors unexpectedly expressed an interest in land, with its demand brusquely rising at an unprecedented pace, especially after the 2008 commodity bubble [2]. According to Deininger [7], in 2009 alone, the demand for land targeting Sub-Saharan Africa, which was fed by a strong LSLAs-component, equaled 20 times its historical average. To get a sense of the global magnitude of this phenomenon, the Land Matrix [8], which is widely recognized as the most comprehensive and up-to-date database on large-scale land transactions, collects information on over 2800 deals since the beginning of the new millennium, corresponding in aggregate to just above 100 million hectares (ha) of land. Two out of three of these large-scale land deals, spreading over 77.3 million ha, are labelled as "*transnational*", while the rest of the records reflect purely domestic land transactions. With a list containing more than 90 destination countries and over 120 investor countries, even acknowledging that the same country can appear both as investor and as targeted region, the global scale of the LSLAs phenomenon is not currently under debate. Since the Land Matrix is constantly updated, it is important to say that these figures were obtained on November 12th, 2018. In addition, the reported figures for the total number of deals and total area includes all records in the dataset, irrespective to their *negotiation status*, according to which a deal can be "*concluded*" (2401 deals, corresponding to 70,347,793 ha), "*intended*" (262 deals over 21,776,190 ha), or "*failed*" (146 deals, equal to 8,620,377 ha).

It is not a coincidence that several empirical papers borrowed from trade economics the (in)famous gravity equation [9]—which already had been borrowed from physics—to analyze different aspects of LSLAs [10–12]. Given this, one of the ways in which we can look at LSLAs is through the lenses of an international market, where land is the main traded commodity. Yet, it seems that this international market for land it is an unusual one, not just because of the peculiar nature of its commodity, which was hardly traded internationally in the past, but also because it appears to be a "*market without prices*" [13]. In this sense, the LSLAs literature pointed out the lack of information and transparency surrounding

the entire life cycle of these investments-from the inception phase, through the negotiations, up to the operational stages [14]. On the one hand, the increasing pressure from civil society, non-governmental organizations (NGOs) and international institutions, in a joint effort to improve the information landscape around LSLAs, is gradually contributing to the disclosure of contractual terms, land records and cadastral registries. Indeed, beside the already mentioned Land Matrix, information on land records can be found on the *Open Land Contract* on-line repository [15], on the *Cadasta* website [16], as well as on the *Land Portal* web platform [17], which contains a rich selection of open access data and information on land tenure and land governance across the globe. On the other hand, it is also true that finding in a systematic way the price at which large-scale land deals are closed is still, to a large extent, a lost cause.

The evidence emerging from the existing LSLAs literature suggests also that high-income and land-scarce countries sought, and are still seeking, land in low-income but land-abundant countries, mainly from the *Global South* [4]. After an initial emphasis put on the *negotiation status*, that is, whether a deal was actually concluded, still under negotiation or cancelled, the second *Analytical Report* from the Land Matrix shifted the attention to the *implementation status*, suggesting that, with more than half of the reported deals currently in operation, the LSLAs phenomenon reached a new "*productive*" era [18].

The LSLAs phenomenon stimulated two main and opposed development narratives. On the one hand, this wave of land-related investment has been hailed as a development opportunity, especially for low- and middle-income countries where agricultural activities are, at the same time, suffering of chronic underinvestment and typically contributing to a large extent to GDP, occupation, and livelihood. On the other hand, these deals have been seen as *land grabbing:* a fierce international competition for the control of natural resources, such as water, forestland, and farmland, which is happening at the expenses of vulnerable local populations [19–22].

From a global perspective, LSLAs can be seen as reflecting the increasing imbalance between the global supply and demand for land, with the "*perfect storm*"—as it was originally defined by Professor Sir John Beddington [23]—in the making [24]: the combined pressure on earth's ecosystems and anthropogenic activities of climate change, population growth, and dietary changes pushing towards higher levels of average daily calorie intake. Such a tremendous combination of factors is likely to exacerbate dramatically the pressure on the planet's food, water, and energy reserves in the next few years, with the risk of disproportionately hitting the poorest and most vulnerable strata of the world's population.

The increasing pressure of this *perfect storm* of human-led factors on natural resources, not only is changing the structure of the land market by making land a globally traded commodity, but it is also inducing a parallel and profound institutional change through the LSLAs phenomenon. In this context, land ownership is increasingly evolving from customary and often collective forms of tenure typically adopted by local communities and indigenous populations to manage common pool resources, towards Western-like forms of individual private property. Indeed, during the last century's decolonization process, only a fraction of the natural resources under traditional forms of tenure was recognized by law, thus making in some cases very difficult to recognize, formalize, and defend local communities' rights associated with customary tenure systems applied to common pool resources, including land. In fact, Dell'Angelo et Al. [20] provide support for the idea that the contemporary rush for land is preferentially targeting areas under traditional and common-property systems, thus embedding a strong "*commons grabbing*" component. According to the authors, while the communities adopting common and customary tenure regimes have developed over time forms of resilience to internal shocks (i.e., other community members), it is not clear to what extent they are equipped to absorb shocks induced by exogenous factors, such as the competition with new external actors for the control over land reserves.

In this context, the crucial role of land clearly emerges, together with the need to allocate land optimally among the increasing number of competing land uses. The reader should note

that "*optimally*", here, is to be intended as the balance—necessarily accounting for the related trade-offs—between economic efficiency, long-term development, inter and intra-generational equity, as well as sustainable management of environmental resources. Therefore, the land governance mechanisms regulating the LSLAs phenomenon, can make the difference—for better or for worse—in facing the *perfect storm*. Yet, in order to implement and coordinate adequate policies, it is important to step back for a moment and start from the understanding of the many faces of land in the contemporary world.

## 3. Methodology

The preliminary phase of this research was structured as a systematic review of the existing LSLAs literature, including scientific books and peer-reviewed journal articles in English. Search terms comprised *LSLAs*, *land grabbing*, but also other locutions and compound forms typically used to refer to this phenomenon, such as *transnational land deals*, *large-scale land investments*, *large-scale land transactions*, *land-based investments*, *foreign land acquisition*, and *land rush*.

This information gathering process, which was also complemented by a press and media review, highlighted the existence of a recent but rapidly growing literature on the topic, covering an extremely rich variety of different, but often interconnected aspects. If the multidimensionality of the LSLAs phenomenon emerged clearly, this review process also highlighted the fact that the economics of LSLAs was, at best, scattered among the different sections of each contribution, revealing the contrast between the terminology used to name the phenomenon itself—which is imbued with economic terms such as *investments*, *deals*, *transactions*—and the lack of a comprehensive economic framework for the understanding of land and the different aspect contributing to its value in the context of LSLAs.

In an effort to disentangle the complexity of the LSLAs phenomenon and to acknowledge its inherent multidisciplinary, I organized this review around four main pillars: *economics*, *space*, *environment*, and *institutions*. These four aspects were broad enough to capture the whole range of features and implications embedded in LSLAs, and at the same time, they proved to be a clear and useful way to differentiate and to organize in a more systematic way the complexity of various bodies of the literature looking at LSALs from different disciplines.

While acknowledging the crucial contribution of other non-economic disciplines to the understanding the multifaceted nature of LSLAs, I decided to address with this research the lack of a conceptual economic framework specifically tailored around land in contemporary LSLAs. I then started to look at how the economic thought addressed other similar land-related issues in the past through the lenses of the four pillars that I had previously identified, combining these elements with a review of the main aspects of LSLAs. The result of this exercise, which is presented in detail in the following sections, is a brief history of land in economics in the context LSLAs.

## 4. Land and Economics

Land appeared in economics at a very early stage of the history of this discipline. Initially it had a prominent role, arguably reflecting the crucial position of agricultural activities in the 18th century society. In his seminal contribution to the economic science, the physiocrat Cantillon [25], who is often seen as the first author to publish a modern economic treaty [26], put land at the center of his theory of value. According to Brems [27], which has the merit of having formalized in a modern and rigorous manner Cantillon's thought, the physiocrat author gave birth to an original "*land theory of value*", where ultimately all production factors were reduced to (indirect) land.

Compared to Cantillon, who was mainly looking at a largely pre-capitalist society, the classical economists were already aware of the rapid ascent of capitalism. They witnessed directly the deep changes that it was producing in the economy and in the society. In this sense, Smith, Ricardo, and Marx gradually began to lose interest in land, while concentrating their efforts in understanding better the novelties of labor and capital within the new capitalistic global order. Nonetheless, Ricardo [28]

introduced the idea that the fertility of land—and thus its productivity—is the main determinant of the agricultural land rent, therefore driving the economic value of this resource.

The simple intuition behind the so-called "*Ricardian approach*" to land—suggesting that the land value can be measured in terms of its agricultural productivity, using for instance measures such as the average or net revenue per hectare for specific crops—is still used and debated at present days. For instance, in an article that appeared in 1994 on the *American Economic Review*, Mendelsohn, Nordhaus, and Shaw assessed the impact of climate change on US agriculture looking at farmland prices and farm revenues, explicitly acknowledging their work as based on a "*Ricardian approach*" [29] (p. 755):

"In this study, we develop a new technique that in principle can correct for the bias in the production-function technique by using economic data on the value of land. We call this the Ricardian approach, in which, instead of studying yields of specific crops, we examine how climate in different places affects the net rent or value of farmland. By directly measuring farm prices or revenues, we account for the direct impacts of climate on yields of different crops as well as the indirect substitution of different inputs, introduction of different activities, and other potential adaptations to different climates. If markets are functioning properly, the Ricardian approach will allow us to measure the economic value of different activities and therefore to verify whether the economic impacts implied by the production-function approach are reproduced in the field"

While having some advantages, including the possibility to account for farmers' adaptation to climate change with relatively less data compared to other methods, the *Ricardian approach* has a strong limitation: it assumes that the land market, together with other related markets, are operating in perfect conditions. If this assumption might stand—at least to some extent—in land and real estate markets in the most advanced economies, it is hard to think that it can reflect the reality in most of developing countries, which also happen to be the most targeted areas by LSLAs. In addition to this, Timmins [30] argued that the application of Ricardian techniques can lead to biased results, especially in the presence of "unobservable determinants of land value" [ibid., p.120] varying across different possible land uses. Therefore, it is not a coincidence that Timmins, as opposed to the US agriculture case studied by Mendelsohn and his colleagues [28], decided to focus his empirical work on Brazil, a complex and vast developing country, rich in both "unobservable determinants of land values" and alternative land uses.

Surprisingly, the question of the (correct) determination of land value appears to have been somehow avoided by LSLAs literature so far. Besides few scattered references to the price of land allegedly paid in one large-scale land deal or another, only the literature investigating the issue of fair compensation for local populations and indigenous communities affected by LSLAs, expressed a more systematic interest in the land value issue [31]. For instance, in his essay addressing fair compensation in transnational land deals, De Maria [32] discussed the controversial aspects related to the correct determination of land value using a law and economics perspective. Among the other cases revised in this essay, the discussion around the famous *Timber Creek Case* (See *Griffiths v. Northern Territory of Australia* (2016) FCA 900, as well as the appeal decision *Northern Territory of Australia v. Griffiths* (2017) FCAFC 106.) is of a particular interest in this context. In the appeal decision, The Full Federal Court of Australia downsized from AUD 3.3 million to 2.9 million the amount of compensation to be paid to the *Ngaliwurru–Nungali* aboriginal people, who were stripped out of their customary land during the development of the town of Timber Creek. However, despite finding mistakes in the calculation of the economic loss suffered by the aboriginal landholders, the Federal Court did not contend the inclusion of non-economics elements—mainly motivated through the spiritual value attached to land by the natives—in the calculation of the final value of the compensation. Despite being at the intersection of law and economics, this case clearly shows the importance of the previously mentioned "*unobservable determinants of land values*".

Trade is another branch of economics that can be of help in understanding LSLAs. Yet, just like the economic valuation of the land value, it probably requires some degree of conceptual adaptation in order to fully capture LSLAs. Indeed, trade economics traditionally assumed land to be a fixed

production factor, both in terms of quantity supplied and mobility, as opposed to capital and labor, which are considered to be, at least to some extent, mobile factors [33,34]. For instance, Kenen [35], who, to be fair, put the emphasis of his work on capital rather than on labor and land, defined land as "*fixed stock [ … ] wholly inert*" [Ibid., p. 441]. In line with this conception, within the most famous and influential models for international trade [36–41], land, when included in the analysis, was considered at best an ancillary production factor, with the focus put mainly on capital and labor. Yet, the recent wave of transnational *Large-Scale Land Acquisitions* (LSLAs) proves that the ownership of land is becoming increasingly mobile, so that each country's endowment of land is not constrained anymore to national borders.

With the *perfect storm* described in the previous section in the background and considering the main LSLAs features, land cannot be considered anymore simply as a stylized, abstract, and fixed production factor. Moreover, if the price of land is still largely a missing element from the LSLAs literature, the *Timber Creek* case shows that the market value of land, alone, is not sufficient for a complete assessment of the value of this resource, which ultimately should include also a variety of non-economic factors. Back in 1944, Polanyi [42] already understood the limitations of a conceptualization of land solely based on its economic functions [ibid., p. 178]:

"The economic function is but one of many vital functions of land. It invests man's life with stability; it is the site of his habitation; it is a condition of his physical safety; it is the landscape and the seasons. We might as well imagine his being born without hands and feet as carrying on his life without land."

If underestimating the importance of the economic functions of land would be a terrible mistake, at the same time we need to acknowledge that land is much more than an economic asset: land is a complex commodity, with both market and non-market features; it supports the livelihood of billions of human beings; it is strategic for feeding the world population; it is a fundamental brick in the architecture of ecosystems and a vital element for building communities resilient to climate change; it is often the ground on which social, cultural, and individual identity are built. The following sections of this paper try to shed some lights on these complex and intertwined aspects, which economists often refer to as *externalities*, and which, nevertheless, concur to define land and its value.

## 5. Land and Space

Urban economics and economic geography represent other branches of the economic science which gave a particular attention to land. The pioneering contribution of Von Thunen [43] (The Von Thünen model was firstly published in 1826, but it seemed more suitable to cite the 1966 critical edition and translation by Hall (ed.) and Wartenberg (transl.), which, to the best of my knowledge, is the first English translation of this incredibly influential work) emphasized the importance of space. In particular, in the original formulation of his model, the author suggested that the land rent, which depends on the level of farm specialization and on the specific land use, is ultimately an inverse function of the distance from the town center. Almost a century and a half later, Von Thunen's land rent theory inspired one of the most influential models in modern urban economics. Indeed, Alonso's *monocentric city model*, constituted the milestone for urban economics for several decades [44]. In his formulation, direct land consumption was the main engine for urban expansion, with the land value ultimately depending on individual preferences over location and on the distance from the so-called (and unique) *Central Business District* (CBD). In the subsequent evolutions of the monocentric city model [45,46], despite remaining an important factor, land was demoted to an "*intermediate input in the production of housing, which is the final consumption good*" [47] (p. 821). In simple words, the great contribution of this family of models relies on the formalization of transport costs, that is, mainly seen as the opportunity cost arising from the distance from the city center, as a determinant of land value in urban context.

If distance matters, also the other factors related to the specific location in space are important. Indeed, Paul Krugman, to whom is credited the original merit of giving birth to the branch of economics

known as *new economic geography*, emphasized the role of space and distance in industrial location choices and trade [48]. The details of this new conception will not be discussed further, because they go beyond the general purpose of this research. Yet, using Krugman's own words, I would like to contextualize the importance and the novelty of this new approach for the general evolution of the economic thought [48] (p. 1): "in the late 1980s mainstream economists were almost literally oblivious to the fact that economies aren't dimensionless points in space, and to what the spatial dimension of the economy had to say about the nature of economic forces."

The importance of space and distance was also highlighted in other trade models, including the aforementioned family of gravity models for trade, from which a series of more recent empirical works took inspiration to understand the forces at play in LSLAs [10–13,49]. Gravity models for LSLAs help to understand the structure and the functioning of the global land market and they reflect the variety of factors and actors, each with his or her own goal, involved in this market. In such a peculiar, complex, and imperfect market—that is, a "*market without prices*" and with very few binding national and international regulations—it is hard to expect that Smith's *invisible hand* is at work. Instead, it is probably easier to see the Marxian critique of the invisible hand at work [50] (p. 1): "Marx's critique of the 'invisible hand' concept does not dwell essentially on the analysis of how a market economy actually operates. It would above all insist that this operation is not eternal, not immanent in 'human nature', but created by specific historical circumstances, a product of a special way of social organisation, and due to disappear at some stage of historical evolution as it appeared during a previous stage. And it would also stress that this 'invisible hand' leads neither to the maximum of economic growth nor to the optimum of human wellbeing for the greatest number of individuals."

In this sense, the expected outcome of LSLAs is not a perfect nor an equitable allocation of the increasingly scarce global land reserves, but it is more the result of the interaction of different—public and private, individual and collective, domestic and foreigner—stakeholders, with preferences and consequences varying, among other factors, with space and geography too.

In general, what we can learn from the past literature when looking at the contemporary land rush, is that the geographical and spatial features of this phenomenon matter. If this intuition is not new in the existing LSLAs literature [18,51–53], the recent and rapid technological developments in remote sensing, satellite imagery, community-based Geographic Information Systems (GIS), mobile-based and drone-based mapping have just started to be systematically applied to the LSLAs context [54]. Indeed, if the global geography of *land grabbing*, together with the distance (or proximity) of investor and destination countries, constitute elements that have been already addressed by several authors and from different angles, the spatial boundaries and features of specific large-scale land deals and concessions, which are also extremely relevant, are often hard to find. In this sense, I wish to stress the current imbalance between the macro-geography and the micro-geography of LSALs, hoping that this consideration will stimulate further research aiming at reducing this gap.

## 6. Land and Environment

Environmental economics questioned since its early stages the ancillary and stylized conception of natural resources that was often embedded in other branches of the economic thought. The publication of "*The Limit to Growth*" in 1972 [55], can be seen as the symbolic moment in which economists realized that they could not ignore anymore the physical, chemical, and biological attributes of the different forms of natural capital, as well as the fact that there were limits to its substitutability. It is not a coincidence that, in the same period, Georgescu-Roegen bridged the gap between economics and physics, thus bringing the economists back to the harsh reality of the fundamental laws of thermodynamic [56].

Environmental economists also stressed the importance of externalities, inspired by the seminal contribution of Pigou [57]. The monetary evaluation of goods that do not necessarily have their own market became a fascinating issue, eventually leading to the awareness that intangible non-market values contribute to the determination of extremely tangible, and sometimes marketable, outcomes,

especially when considering environmental resources. Gradually, the economic sciences embraced concepts we are all now familiar with, such as pollution, biodiversity, natural resource management, sustainability, and climate change.

Among others, the issues related with climate change received particular attention in the last decades, producing a tremendous acceleration in land-use modelling techniques. For instance, Hertel, Rose, and Tol [58] edited an entire volume describing the myriad of existing models for land-use and land-use change developed over the last thirty years or so. More recently, the Intergovernmental Panel on Climate Change (IPCC) *Special Report on Global Warming of 1.5°* highlighted how the global land reserves are already deeply affected by climate change, and, at the same time, how land management represent a crucial component in the mix of proposed responses [59]. Land is so important in this context that the IPCC also plan to release a new *Special Report on Climate Change and Land* (SRCCL) next year.

With an estimated 65% of global land reserves *de facto* held by indigenous and local communities under customary and often collective tenure regimes [60], typically adopting small-scale and low-intensity techniques of agricultural, fishery, and forestry production, the current interest of international investors in LSLAs is both, *literally* and *figuratively* changing the landscape. Scholars are already aware of the potential for land-use change and environmental impact embedded in LSLAs [19,61,62]. International institutions, civil society, and the private sector are aware of the potential impacts of LSLAs too, as suggested by the increasing number of guidelines, protocols, and tools—such as the FAO-endorsed *Voluntary Guidelines on the Responsible Governance of Tenure of Land, Fisheries and Forests in the Context of National Food Security* [63]—to promote responsible and sustainable land investments. However, a more rigorous understanding and quantification of the effects of the current proliferation of large-scale land investments on biodiversity, climate changes, and land-use is still needed. In other words, the LSLAs literature has just started to disentangle the implications around the agro-ecological and pedoclimatic features of the large strips of land that are currently being acquired and sold.

## 7. Land and Institutions

The debate around (formal and informal) institutions, growth, and development—that is, whether good institutions are the major cause of economic growth and human development, or, conversely, high level of accumulation of human and social capital are actually responsible for improvements in institutional quality—is still open [64–68]. However, all opposed factions agree that the issues of property rights and tenure systems, standing at the core of many aspects of land-related research, are crucial in this context [69].

Land is not just a good defined by its economic rent, its position in the space and its natural features, but it is also a political, social, spiritual, and cultural asset. Land is so deeply embodied in the collective imagination of many societies, that it contributes, among other functions, to define the social identity both at the individual and at the collective level. Indeed, according to Deininger and Feder [70] (p. 1):

"The way in which land rights are assigned therefore determines households' ability to produce their subsistence and generate marketable surplus, their social and economic status (and in many cases their collective identity), their incentive to exert non-observable effort and make investments, and often also their ability to access financial markets or to arrange for smoothing of consumption and income."

The "*institutional superstructure*" attached to land, that is, the way in which social customs and official legal systems allocate property rights and regulate access and use of land, it is not static and evolves within time and space. The historical evidence suggests that the actual path that this evolution takes can deeply affect the evolution of societies themselves. For instance, the *enclosures* changed not just the landscape in the United Kingdom during the period 1750–1850, but they also sanctioned the passage from a pre-capitalist rural society to a capitalist and industrial one [71,72]. In more recent

days, institutional arrangements over land are not less important than they were during the British industrial revolution, and the actual shape they take can range over an incredibly vast horizon of different possibilities. Indeed, according to Feder and Feeny [73] (p. 135):

"[ . . . ] in the contemporary world, especially in developing countries, the presumption of exclusive, transferable, alienable, and enforceable rights is frequently inaccurate and potentially misleading. In such cases the complex nature of institutional arrangements in general and property rights in particular needs to be described".

In this sense, between the end of the last century and the beginning of the present one, the Nobel laureate Elinor Ostrom has laid the foundations for unravelling the knot linking the diverse range of property rights, the variety of existing formal and informal institutions, and their relation with natural resources [74,75].

However, when it comes to LSLAs, the knot seems to be not fully unraveled yet. Indeed, according to the Rights and Resources Initiative (RRI), more than half of the global land reserves are held by indigenous people and communities under a diverse array of customary tenure regimes, but their ownership is formally recognized only on one tenth of the global land surface [60].

Several authors argued that this lack of formal recognition and enforcement of traditional tenure systems, can increase the risk of *land grabbing*, while reducing the room for LSLAs to create inclusive and tangible development opportunities [1,3,76]. Other authors highlighted also that the impact of specific large-scale land deals on affected communities, depending on different forms of traditional and customary land rights, can be very diverse among the various subgroups within each community, such as women, youths or elites [77–79]. Interestingly, customary tenure regimes are not only a trigger for land grabbing, but they can work also as a local community response, that is, a mechanism of social resilience, to transnational LSLAs [80].

The previously cited LSLAs gravity literature also emphasized the role of institutions, almost unanimously recognizing tenure insecurity as one of the main drivers for large-scale land investments [10,12,13,49]. The latest findings in this branch of the quantitative LSLAs research, unveil an extra layer of complexity surrounding the interplay between foreign land acquisition and institutions. At the same time, these findings reinforce the idea proposed in this article, where I suggest that a more holistic conceptualization of land is needed to fully understand the implications of LSLAs. Indeed, Raimondi and Scoppola [11], not only address the issue of the institutional distance between target and investor country, but also find that the institutional pattern changes with the geography of LSALs [Ibid., p. 537]:

"The hypothesis that Africa follows a clearly different pattern from other regions is confirmed by the results. Indeed, while political distance negatively affects FLA, the gap in governance fosters the amount of hectares acquired in Africa, though not the number of contracts. These results suggest that the weaker the level of governance in target countries in Africa, the more investors prefer large-scale contracts".

## 8. Conclusions

Before entering the maze of the qualitative and quantitative impacts of LSLAs and before discussing land governance and policy implications, it is important to take a step back, understanding first the essence of the multiple and intertwined dimensions behind land in the context of the contemporary wave of LSLAs. To do so, I organized this review around four aspects-namely *economics*, *geography*, *environment*, and *institutions*, that proved to be a practical way to categorize, to order, and to connect different LSLAs features with the range of economic theories that could contribute to their understanding. Indeed, the value of land is not only about the pure economics of it, but it is also about its location in the space, its environmental and pedoclimatic features, and the variety of both formal and informal institutions that contribute to land governance in different societies. This critical review exercise suggested that many of the elements that appear as original features of current LSLAs are actually not new. Therefore, looking at how they have been conceptualized and approached in the

past, offered valuable insights for the comprehension of the present rush for land and stimulated a critical reflection on how the LSLAs-related research could be improved by adopting a more holistic approach over land issues.

The first important finding that emerged from this approach is in fact the existence of a gap in the LSLAs literature. Indeed, in the previous sections of this paper I argued how the LSLAs phenomenon embodies new trajectory in the contemporary process of land *commodification*. Yet, with a new international market for land in place and with a constant reference to the economics of LSLAs rooted in its definition and in the associated vocabulary, it is astonishing how the discourse around the value of land is systematically missing from the LSLAs-related literature. If the lack of transparency surrounding LSLAs and the following shortage of reliable data over land prices can contribute to explain this situation, it is also true that the contemporary economic science appears not to be equipped with a holistic theoretical framework for land, allowing for the full understanding of the implications of the current surge in transnational land acquisitions.

In this sense, while highlighting the need for additional and more rigorous economic analyses in the context of LSLAs, the historical perspective adopted in this paper suggests that international trade economics offers a preliminary way to include in the analysis some of the peculiar aspects of LSLAs, namely the transnational nature of LSLAs and the increased mobility of land—and its ownership—as a production factor. The review of the intertwined relationship between land and economics also emphasized how the *Ricardian approach* to land value, according to which the value of land ultimately depends on its productivity, with such value being fully captured by existing land markets, would not take into account all the complex and peculiar aspects characterizing LSLAs and, more generally, land in the 21st century.

This last consideration leads us to the second key finding of this paper, that is the need to go beyond the silos of different schools of thought in economics, as well as to rely also on the approaches, the theories, and the tools provided by other disciplines outside the fields of economics and development studies. The inclusion of the sections on space, environment, and institutions alongside the one on land and economics, can be seen as a preliminary step in this direction. The lesson learned here is that LSLAs is not happening in a spatial, environmental, and institutional vacuum. Different aspects of the geography of these deals should be factored into LSLAs studies, in an effort to understand better the implications of both the physical distance-for instance in terms of location preferences, transport costs, and transaction costs-and the socio-institutional distance-for instance in terms of the diversity of tenure regimes, land laws, and customs among the actors involved. This review also stresses that LSLAs are happening within natural ecosystems, with implications on biodiversity, pollution, and climate change that still need to be explored more in depth. In addition, economic, spatial, environmental, and institutional aspects influencing LSLAs and the value of land are not static: they evolve over time together with the actors involved and their motives. For instance, an historical perspective over colonization and decolonization dynamics occurred over the last century can help understanding the players and the features of the current LSLAs. Similarly, a brief history of land in economics and in related sciences can help the understanding of the factors determining the value of land and the likely outcome that we will observe in the future on the newly born international market for land.

The recent wave of LSLAs suggests that land is indeed a commodity, but together with its economic functions, it also shows the variety of other elements that ultimately contribute to the actual determination of its value. Taking advantage of the variety of different approaches proposed over time in the history of economic thought and other related disciplines, as well as assessing them in the light of the current features of LSLAs, this paper ultimately sets the ground for a new holistic conceptualization of land that reflects its present complexity.

**Funding:** This research received no external funding.

**Acknowledgments:** I must acknowledge that most of this work has been carried out during my PhD in Agricultural and Food Economics at the School of Agriculture, Policy, and Development of the University of Reading, where I was awarded a 3-year full Research Studentship in Social Science. I wish to express my deep gratitude to my supervisors, Giacomo Zanello and Elizabeth Robinson, for their professional guidance, enthusiastic encouragement and constructive critiques throughout all phases of this research. I would also like to extend my thanks to a number of colleagues from the University of Reading, from the Land Portal Foundation, as well as from other Institutions, who kindly provided further precious recommendations. Finally, I am grateful to Marcella, for her patience and constant support.

**Conflicts of Interest:** The author declares no conflict of interest.

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
