# Peer review of "Understanding Land in the Context of Large-Scale Land Acquisitions: A Brief History of Land in Economics"

_land, doi:10.3390/land8010015_

Round 1

Reviewer 1 Report

This paper revisits the well-trodden area of the concept of land in the field of economics. The novelty is that it undertakes the analysis in the context of large scale land acquisitions. If anything, the paper presents a timely reminder of the complexity of land, and the need for academic disciplines, and the land sector, to incorporate this complexity into analysis of LSLA. This contribution alone makes the paper worthy of publication, even if the 4 aspects identified as important - namely space, economics, environment and institutions – are perhaps not so revealing.

The paper is well written, structured, and arguably only requires minor updates. First, given the cross-disciplinary nature of the journal, it seems important that the paper more directly reveal the underlying research paradigm and methodologies underpinning the work. Whilst the basis is obviously analysis of economic literature, the specific ontology, if not epistemology, and theoretical perspective could be more clearly revealed. In this vein, the authors might consider the inclusion of methodology section.

Following on, whilst the paper is rooted in economic discussion, the paper is encouraging a more holistic appreciation of the complexity inherent the concept of land. The underpinning literature review cannot be doubted as robust, however, the majority of sources a stem purely from economics and/or development, meaning the similar arguments made in other disciplines, on the need for more holistic cross-cutting appraisals of LSLA are potentially missed. It is not suggested that the paper needs to incorporate these other non-economics bodies of work, but, increased acknowledgement that they exist, and potentially reveals similar arguments as posited in the paper at hand, might be appropriate.

Finally, the authors could go further in the conclusions – or an extended discussion section – to articulate more clearly the implications or potential impact of the results developed and presented here. How can researchers realistically incorporate all 4 of the aspects identified here, into research work and projects? What different understandings of LSLA might be expect? Can we make some hypotheses on these? Posing and attempting to present initial answers to these sorts of questions would further impress the relevance and importance of the work.     

Author Response

I wish to thank the anonymous Reviewer for the stimulating comments that were provided. In an effort to improve this paper, I tried to incorporate in the revised manuscript all the suggestions made and I tracked all changes in word.

This paper revisits the well-trodden area of the concept of land in the field of economics. The novelty is that it undertakes the analysis in the context of large scale land acquisitions. If anything, the paper presents a timely reminder of the complexity of land, and the need for academic disciplines, and the land sector, to incorporate this complexity into analysis of LSLA. This contribution alone makes the paper worthy of publication, even if the 4 aspects identified as important - namely space, economics, environment and institutions – are perhaps not so revealing.

The paper is well written, structured, and arguably only requires minor updates. First, given the cross-disciplinary nature of the journal, it seems important that the paper more directly reveal the underlying research paradigm and methodologies underpinning the work. Whilst the basis is obviously analysis of economic literature, the specific ontology, if not epistemology, and theoretical perspective could be more clearly revealed. In this vein, the authors might consider the inclusion of methodology section.

Thanks for raising this issue. I have added a brief methodology section in the paper and I hope it is in line with the request raised by the Reviewer. I tried to clarify the different steps involved of this research and its theoretical perspective.

Following on, whilst the paper is rooted in economic discussion, the paper is encouraging a more holistic appreciation of the complexity inherent the concept of land. The underpinning literature review cannot be doubted as robust, however, the majority of sources a stem purely from economics and/or development, meaning the similar arguments made in other disciplines, on the need for more holistic cross-cutting appraisals of LSLA are potentially missed. It is not suggested that the paper needs to incorporate these other non-economics bodies of work, but, increased acknowledgement that they exist, and potentially reveals similar arguments as posited in the paper at hand, might be appropriate.

This is a very good point. I tried to incorporate the acknowledgment of the important contribution made by non-economic disciplines both in the new methodology section and in the revised and expanded conclusions. At the same time, I tried to justify more clearly why this specific paper focuses more on economic aspects.

Finally, the authors could go further in the conclusions – or an extended discussion section – to articulate more clearly the implications or potential impact of the results developed and presented here. How can researchers realistically incorporate all 4 of the aspects identified here, into research work and projects? What different understandings of LSLA might be expect? Can we make some hypotheses on these? Posing and attempting to present initial answers to these sorts of questions would further impress the relevance and importance of the work.    

I edited and expanded the conclusions in order the meet this request. I tried to present better the key lessons learned from this review and I tried to delineate possible ways in which LSLAs studies can incorporate these lessons. 

Reviewer 2 Report

I like the paper and the 4 views as worked out.

I find esp. line 18 and 'the wake of lsla' in title promising more than actually is delivered; which is the historical overview on the 4 views, which I like and for me is enough; I suggest you tone done the text on esp. international market etc in abstract

start of introduction, line 39-45 is very generic and broad, not to the point and not needed to bring the message you want to bring, I suggest you start closer to the topic, and if you feel compelled put the first lines a bit later in it.

I miss a reference to the work of G. Schoneveld his phd/papers; which also talks about many aspects of lsla in several settings (economic is weakest though); he appears as 2nd author in one publication

I miss a 

line 47 recent surge, it is more than 10 years in the making, is that still recent ?

line 82 feed --> fed

a number of times, e.g. line 90 your way of arguing feels more like a verbal persuasive style than like academic writing; it makes the paper read nice, but I would not mind toning it done a bit ('is not a question',

line 95 it can go before is an u..

line 151 in terms OF its pro

line 153 in an article THAT appeared (or in an article appearing in)

line 166 I dont now indubitably myself

line 171 most targetED areas ?

line 270 not a case ? would coincidence be more precise ??

line 293 being grabbed - so far you are quite neutral on such terms, why put it strongly here ? (if you did it on purpose I am okay with it)

line 299 that THE issue

lien 303 the imaginary ... of; idea of the like ?

324, the the is double

328 than XXX half, get the a out; also , is not needed I feel

353 in THE previous

Author Response

I wish to thank the Reviewer for the detailed review and for the useful comments. 

I did my best to address all suggestions made by the Reviewer and I described below (in red) all the proposed corrections and edits. 

All changes made to the revised manuscript have been documented also using the "Track changes" function of Word.

I like the paper and the 4 views as worked out.

Thanks! 

I find esp. line 18 and 'the wake of lsla' in title promising more than actually is delivered; which is the historical overview on the 4 views, which I like and for me is enough; I suggest you tone done the text on esp. international market etc in abstract

I modified the title and removed the reference to the international market for land from the abstract

start of introduction, line 39-45 is very generic and broad, not to the point and not needed to bring the message you want to bring, I suggest you start closer to the topic, and if you feel compelled put the first lines a bit later in it.

I removed lines 39-45, which were indeed too generic

I miss a reference to the work of G. Schoneveld his phd/papers; which also talks about many aspects of lsla in several settings (economic is weakest though); he appears as 2nd author in one publication

I added two additional references to the work of Schoneveld in two different sections of the paper (see ref no. 6 and no. 48). References to the work of this author were indeed missing, and I feel they contribute to depict a more complete picture of the relevant literature on LSLAs.

line 47 recent surge, it is more than 10 years in the making, is that still recent ?

I replaced "recent" with "observed in the last decade"

line 82 feed --> fed

Noted and corrected

a number of times, e.g. line 90 your way of arguing feels more like a verbal persuasive style than like academic writing; it makes the paper read nice, but I would not mind toning it done a bit ('is not a question',

I modified "is not of a question". Now it reads "it is not under debate". I couldn't locate other instances in the paper -- beside those highlighted by the Reviewer in this point and in the first one -- where I could use a more academic writing style. Nevertheless, I would be more than happy to address further instances that the Reviewer might want to highlight 

line 95 it can go before is an u..

Noted and corrected

line 151 in terms OF its pro

Noted and corrected

line 153 in an article THAT appeared (or in an article appearing in)

Noted and corrected

line 166 I dont now indubitably myself

Noted and corrected

line 171 most targetED areas ?

Noted and corrected

line 270 not a case ? would coincidence be more precise ??

Noted and corrected

line 293 being grabbed - so far you are quite neutral on such terms, why put it strongly here ? (if you did it on purpose I am okay with it)

Noted and corrected. Now it reads "being acquired"

line 299 that THE issue

Noted and corrected 

lien 303 the imaginary ... of; idea of the like ?

Noted and modified. Now it reads "embodied in the collective imagination of many societies"

324, the the is double

Noted and corrected 

328 than XXX half, get the a out; also , is not needed I feel

Noted and corrected 

353 in THE previous

Noted and corrected